



# The predictive power of ice sheet models and the regional sensitivity of ice loss to basal sliding parameterisations: A case study of Pine Island and Thwaites Glaciers, West Antarctica

Jowan M. Barnes[1] and G. Hilmar Gudmundsson[1]

[1]Department of Geography and Environmental Sciences, Northumbria University, Newcastle upon Tyne, UK

**Correspondence:** Jowan M. Barnes (jowan.barnes@northumbria.ac.uk)

**Abstract.** Ice sheet models use a wide range of sliding laws to define a relationship between ice velocity and basal drag, generally comprising some combination of a Weertman-style power law and Coulomb friction. The exact nature of basal sliding is not known from observational data, making assessment of the suitability of different sliding laws difficult. The question of how much this choice could affect predictions of future ice sheet evolution is an important one. Here we conduct

a model study of a large sector of the West Antarctic Ice Sheet (WAIS), a particularly critical component of the cryosphere, using a range of sliding parameterisations, and provide an assessment of the sensitivity of ice loss to the choice of sliding law. We show that, after initialisation, various sliding laws result in broadly similar ranges of sea-level contribution over 100 years, with the range primarily dependent on exact parameter values used in each sliding law. Comparing mass loss from Thwaites and Pine Island glaciers, and the neighbouring regions, reveals significant qualitative geographical differences in the

relationship between sliding parameters and the modelled response to changes in forcing. We show that the responses do not necessarily follow universal systematic patterns and, in particular, higher values of the sliding exponent $m$ do not necessarily imply larger rates of mass loss. Despite differences in the magnitudes of ice loss and rates of change in the system, all our experiments display broad similarities in behaviour which serve to reinforce the decade-to-century-scale predictive power of ice sheet models, regardless of the choice of basal sliding.

## 1    Introduction

For fast flowing areas of the Antarctic Ice Sheet, a significant portion of the forward motion is due to sliding along the bedrock beneath them. Ice sheet models must mathematically represent this sliding in order to simulate the physical system, but the exact nature of the relationship between a glacier and its bedrock cannot be determined from observational data, and is not

fully understood. This process enters models in the form of a sliding law describing the form of basal drag. In general, these laws express basal drag in terms of the basal velocity and/or effective pressure. Sliding laws contain some parameters whose values are unknown, and are prescribed as part of setting up a model. As a result, they can vary between models, and between





simulations, giving different results in future predictions. While there are some common choices for these parameters, it is
unknown which are the most representative of the physics of the ice sheet system.


An early sliding law, and one which is still widely used in modelling, is that of Weertman (1957). This is a power law of the
form

$$\boldsymbol{\tau}_b = C\boldsymbol{v}_b^m, \tag{1}$$

where $\boldsymbol{\tau}_b$ is the basal drag, $C$ is a sliding parameter, $\boldsymbol{v}_b$ is the basal velocity and $m$ is the sliding exponent. We refer to this as
the Weertman sliding law.

Throughout its history, the Weertman law has been criticised for neglecting the effects of cavitation, in which water under
the ice can affect the basal drag. Lliboutry (1968) proposed the idea that a sliding law should depend on effective pressure, the
difference between the hydrostatic overburden of ice and the basal water pressure. This can be done using Coulomb's law of
solid friction, giving the relationship

$$\boldsymbol{\tau}_b = \mu_k N, \tag{2}$$

where $\mu_k$ is a coefficient of friction and $N$ is the effective pressure.

The idea of including effective pressure in the sliding law has been explored further. Budd et al. (1979) proposed a law of
the form

$$\boldsymbol{\tau}_b = N^{\frac{q}{m}} C_B \boldsymbol{v}_b^m, \tag{3}$$

where $C_B$ is a sliding parameter and $q$ is a positive constant. Fowler (1987) produced a sliding law with broadly similar be-
haviour to that of Budd et al. (1979), but which approached Coulomb friction at high velocities. Schoof (2005) asserts that
laws taking the form of Equation 3 are unphysical as they allow arbitrarily large stresses at the bed. Motivated by the work of
Iken (1981), Schoof (2005) derived a regularised Coulomb sliding law, stated to be "essentially the same" as that of Fowler
(1987). Variations of this have been used in several models since (e.g., Gagliardini et al., 2007; Leguy et al., 2014; Cornford
et al., 2020).

Another approach, suggested by Tsai et al. (2015), is to compute the basal drag using both a Weertman sliding law and
a Coulomb friction law, and take the minimum value of the two at each point. We refer to this as the Tsai sliding law. The
Coulomb friction is applicable to softer beds, and is more representative of sliding near the grounding line. Taking the mini-
mum value of this and the Weertman law effectively applies Coulomb friction near the grounding line, and Weertman sliding
elsewhere.



With several sliding laws available, and parameter options within them, it is useful for modellers to know more about the possible implications of their choice. Previous studies have compared sliding laws in different ways. Brondex et al. (2017) compares the Weertman, Budd, regularised Coulomb and Tsai sliding laws in an idealised setting, finding different levels of transient response in each case. This was followed up in Brondex et al. (2019) by simulations on an ASE domain using linear and nonlinear versions of the Weertman and Budd laws, and two friction coefficients in the regularised Coulomb law. In both
studies, the Weertman law was found to produce the smallest changes in volume above flotation (VAF), while the Budd law caused the largest changes. Simulations using the non-linear versions of the Budd and Weertman sliding laws lost ice faster than those using the linear versions.

The effect of the value of $m$ in the Weertman sliding law has been investigated in a few cases. Gillet-Chaulet et al. (2016)
tests different values, and finds that higher values of $m$ generally produce higher velocities and a closer match to an observed surface velocity field of Pine Island Glacier (PIG), with $m = 20$ suggested as the best value in this case. Joughin et al. (2019) compares experiments using the Weertman law with different values of $m$ to a regularised Coulomb law, focusing on a profile along PIG. The regularised Coulomb law was found to reproduce observations with greater success, and higher values of $m$ in the Weertman law approached a better result. De Rydt et al. (2021) explores the idea that $m$ in the Weertman sliding law
does not necessarily need to take a uniform value, presenting a map of "optimal" $m$ values over PIG which come closest to reproducing observed velocity changes. In general, this involves higher values of $m$ on the fastest flowing areas. Sun et al. (2020) compares the response of several models to ice shelf collapse. It is shown that there is a general tendency for models using higher values of $m$ to respond faster to the changes in forcing. This is in agreement with results from within single models in Brondex et al. (2019), Joughin et al. (2019) and De Rydt et al. (2021).

Some examples can be found of the opposite being true, indicating a more complex relationship between the value of $m$ and the rates of change to a system. For Pine Island Glacier in particular, Nias et al. (2018) shows that higher values of $m$ can produce higher or lower changes in VAF depending on the bathymetry product being used. This sensitivity to topography is further demonstrated by Wernecke et al. (2022).

The focus of the work presented in this paper is on the effects of parameter choices in sliding laws on the forward evolution of an ice sheet system and the response to changes in ice shelf forcing. The values of $m$, $\mu_k$ and $q$ are varied, and we seek to quantify the difference these choices can make to the modelled response when different changes in the buttressing of an ice sheet system are applied, as well as whether there are universal systematic patterns when varying sliding parameters. We
also set out to assess the degree of confidence with which predictions can be made by an ice sheet model, despite the lack of empirical knowledge of basal conditions to inform a sliding law.

The chosen domain for the experiments presented in this paper is the Amundsen Sea Embayment (ASE) in West Antarctica (Figure 1), the region of the continent which is experiencing the fastest ice mass loss (Shepherd et al., 2018; Rignot et al.,





2019), at a rate which is accelerating (Sutterley et al., 2014). Understanding how the West Antarctic Ice Sheet (WAIS) is evolving is seen as one of the most important questions in Antarctic research (National Academies of Sciences, Engineering, and Medicine, 2015). A lot of past work has been centred around PIG, including the previously mentioned studies. Recently, there has also been a focus on Thwaites Glacier due to the rapid changes observed in recent years (e.g., Mouginot et al., 2014; Milillo et al., 2019), and it is the subject of a large multidisciplinary investigation, the International Thwaites Glacier Collaboration

(Scambos et al., 2017). Many models have been used to make predictions of the future evolution of Thwaites Glacier and the ASE region, and they generally agree that the trend of rapid retreat in the ASE will continue into the future (e.g., Favier et al., 2014; Joughin et al., 2014; DeConto and Pollard, 2016; Yu et al., 2018). Models tend to differ on the rates of change they calculate, for a variety of reasons (e.g., Cornford et al., 2020), and as such efforts to improve the understanding and functionality of all aspects of ice sheet models, and to reduce uncertainty in their predictions, are ongoing.


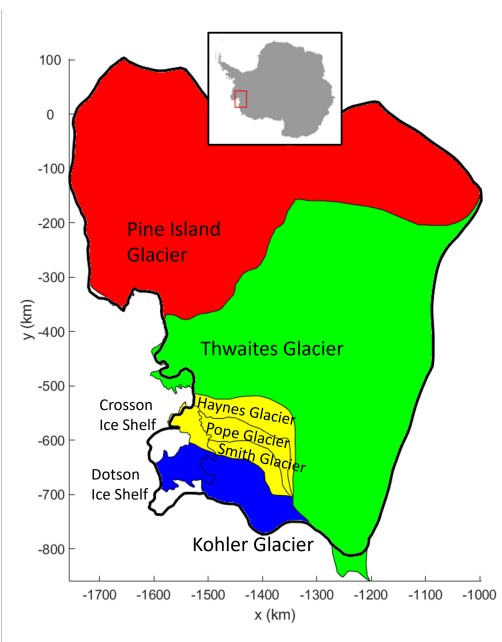

**Figure 1.** The catchment basins of the Amundsen Sea Embayment (ASE) using MEaSUREs Antarctic Boundaries (Rignot et al., 2013) and, inset, the location of the ASE within Antarctica. The different colours indicate the four regions we focus on. The thicker black outline is the boundary of our model domain.





## 2 Experimental design

### 2.1 Model setup and forcing

We use the ice sheet model Úa (Gudmundsson, 2020), which implements the Shallow Shelf Approximation (SSA), and Glen's flow law (Glen, 1958),

$$\dot{\epsilon} = A\boldsymbol{\tau}^n, \tag{4}$$

where $\dot{\epsilon}$ is the strain rate, $A$ is a flow rate factor, $\boldsymbol{\tau}$ is the deviatoric stress and $n$ is a flow law exponent, which we fix at $n = 3$. Our domain covers the ASE region including PIG, Thwaites Glacier and the glaciers flowing into the Dotson and Crosson ice shelves. The inland boundary was created based on the MEaSUREs Antarctic Boundaries (Rignot et al., 2013). We use the geometry from BedMachine Antarctica v2 (Morlighem et al., 2020), from which we derive the calving front boundary for our domain. The mesh resolution is $1.5\,\mathrm{km}$ at the grounding line, and becomes coarser further upstream.

The densities are given the values of $917\,\mathrm{kg\,m^{-3}}$ for ice and $1027\,\mathrm{kg\,m^{-3}}$ for ocean water. The surface mass balance is taken from RACMO2.3p2 (Van Wessem et al., 2018). Where the ice is afloat, we use a simple depth-based melt parameterisation of ocean-induced melt

$$M = \begin{cases} 0 & \text{if } z \geq 0 \\ -\frac{M_{\max}}{500} z & \text{if } 0 > z > -500 \\ M_{\max} & \text{if } z \leq -500, \end{cases} \tag{5}$$

where $M$ is the basal melt rate in $\mathrm{m\,a^{-1}}$, $M_{\max}$ is a prescribed maximum melt rate and $z$ is the vertical coordinate in metres, positive upwards with zero at sea level.

Our experiments test four different sliding laws, and vary the parameters within them. We use the Weertman, Budd and Tsai sliding laws introduced in section 1, and a regularised Coulomb law, which is a variation of the sliding law of Schoof (2005) as used in Cornford et al. (2020), a power-weighted reciprocal sum. We present these sliding laws below in a form which makes clear how they relate to each other.

$$\text{Coulomb friction:} \qquad \boldsymbol{\tau}_C \quad = \quad \mu_k N \frac{\boldsymbol{v}_b}{\|\boldsymbol{v}_b\|} \tag{6}$$

$$\text{Weertman sliding:} \qquad \boldsymbol{\tau}_W \quad = \quad \beta^2 \boldsymbol{v}_b \tag{7}$$

$$\text{Budd sliding:} \qquad \boldsymbol{\tau}_B \quad = \quad N^{\frac{q}{m}} \beta_B^2 \boldsymbol{v}_b \tag{8}$$

$$\text{Tsai sliding:} \qquad \boldsymbol{\tau}_T \quad = \quad \min(\boldsymbol{\tau}_C, \boldsymbol{\tau}_W) \tag{9}$$

$$\text{Regularised Coulomb sliding:} \qquad \frac{1}{\boldsymbol{\tau}_R^m} \quad = \quad \frac{1}{\boldsymbol{\tau}_W^m} + \frac{1}{\boldsymbol{\tau}_C^m} \quad \Longrightarrow \quad \boldsymbol{\tau}_R \quad = \quad \frac{\mu_k N \beta^2 \boldsymbol{v}_b}{((\mu_k N)^m + (\beta^2 \|\boldsymbol{v}_b\|)^m)^{\frac{1}{m}}}, \tag{10}$$



where $\beta^2 = C^{-\frac{1}{m}} \|\boldsymbol{v}_b\|^{\frac{1}{m}-1}$. Note that Coulomb friction alone is not used in any of our model simulations. The parameters are given a range of values, exploring a range around common values used in modelling studies. We use $m = 1, 3, 5, 7$ and $9$. Commonly, $m = 3$ is used by ice sheet models, but higher values of $m$ could produce behaviour closer to that of regularised Coulomb laws. We use $q = 1$ and $2$. The value $q = 2$ was originally suggested by Budd et al. (1984), but $q = 1$ is a more common choice in recent studies (e.g., Brondex et al., 2019). A range of friction coefficients have been used in modelling, and mostly fall within the range of our chosen values, $\mu_k = 0.25, 0.5, 0.75$.

## 2.2 Overview of experiments

We use an adjoint inversion method to calculate fields of values for $A$ and $C$ independently for each set of parameters in each sliding law. The method involves referring to observed velocities $u_{\mathrm{obs}}$ and $v_{\mathrm{obs}}$ with observational errors $u_{\mathrm{err}}$ and $v_{\mathrm{err}}$, from the 2016-17 year of the dataset presented by Mouginot et al. (2017), to minimise a cost function $\mathcal{J} = I + R$, consisting of a misfit function

$$I = \frac{1}{2\mathcal{A}} \int ((u - u_{\mathrm{obs}})/u_{\mathrm{err}})^2 \, \mathrm{d}\mathcal{A} + \frac{1}{2\mathcal{A}} \int ((v - v_{\mathrm{obs}})/v_{\mathrm{err}})^2 \, \mathrm{d}\mathcal{A}, \tag{11}$$

and a regularisation term

$$R = \sum_{k=1,2} \frac{1}{2\mathcal{A}} \int \left( \gamma_{s,k}^2 (\nabla(p_k - p_{k,\mathrm{prior}}))^2 + \gamma_{a,k}^2 (p_k - p_{k,\mathrm{prior}})^2 \right) \mathrm{d}\mathcal{A}, \tag{12}$$

where $\mathcal{A} = \int \mathrm{d}\mathcal{A}$ is the total area, $u$ and $v$ are the modelled horizontal $x$ and $y$ velocity components, respectively, $\gamma_{s,k}$ and $\gamma_{a,k}$ are slope and amplitude regularisation parameters, $p_1 = \log_{10} A$, $p_2 = \log_{10} C$ and $p_{k,\mathrm{prior}}$ are prior values, or initial estimates, for the parameters $p_k$. For the inversions in this study, $\gamma_{s,1} = \gamma_{s,2} = 1 \times 10^4$ m, $\gamma_{a,1} = 100$ and $\gamma_{a,2} = 1$. Further details of the inversion method can be found in Barnes et al. (2021).

Each simulation is run for 20 years using $M_{\mathrm{max}} = 100$. This allows for any initial adjustments in the model to settle, and the system to start evolving steadily. We do not introduce different forcing from the beginning of the run, as we wish to assess the effects of a change introduced to a system which is already in motion. Changes are introduced after 20 years of simulation, and the experiments are split into four branches from this point:

- *Control Branch* - the simulation continues with the same forcing.

- *Increased Melt Branch* - there is an immediate step change increase in the melt rate, with the value of $M_{\mathrm{max}}$ raised to 200.

- *Decreased Melt Branch* - melting stops immediately, with $M_{\mathrm{max}}$ lowered to 0.

- *Ice Shelf Removed Branch* - all the floating ice is immediately removed and no more is allowed to form for the remainder of the run.



These four branches are run for a further 100 years. These are not realistic perturbations, but are designed to test the most extreme changes to the ice shelf buttressing, as well as one more moderate case, in order to clearly see the differences caused
by altering the sliding parameterisation.

## 3   Results

The experiments in the *Control Branch* all display loss of ice from the system, with the loss of volume above flotation (VAF) over the 120 years of simulation being between 37 and 104 mm sea level equivalent, depending on the sliding law and param-
eter choices (Figure 2 and Table 1). The mean ice loss is 59.5 mm sea level equivalent. There is a clear pattern of lower values

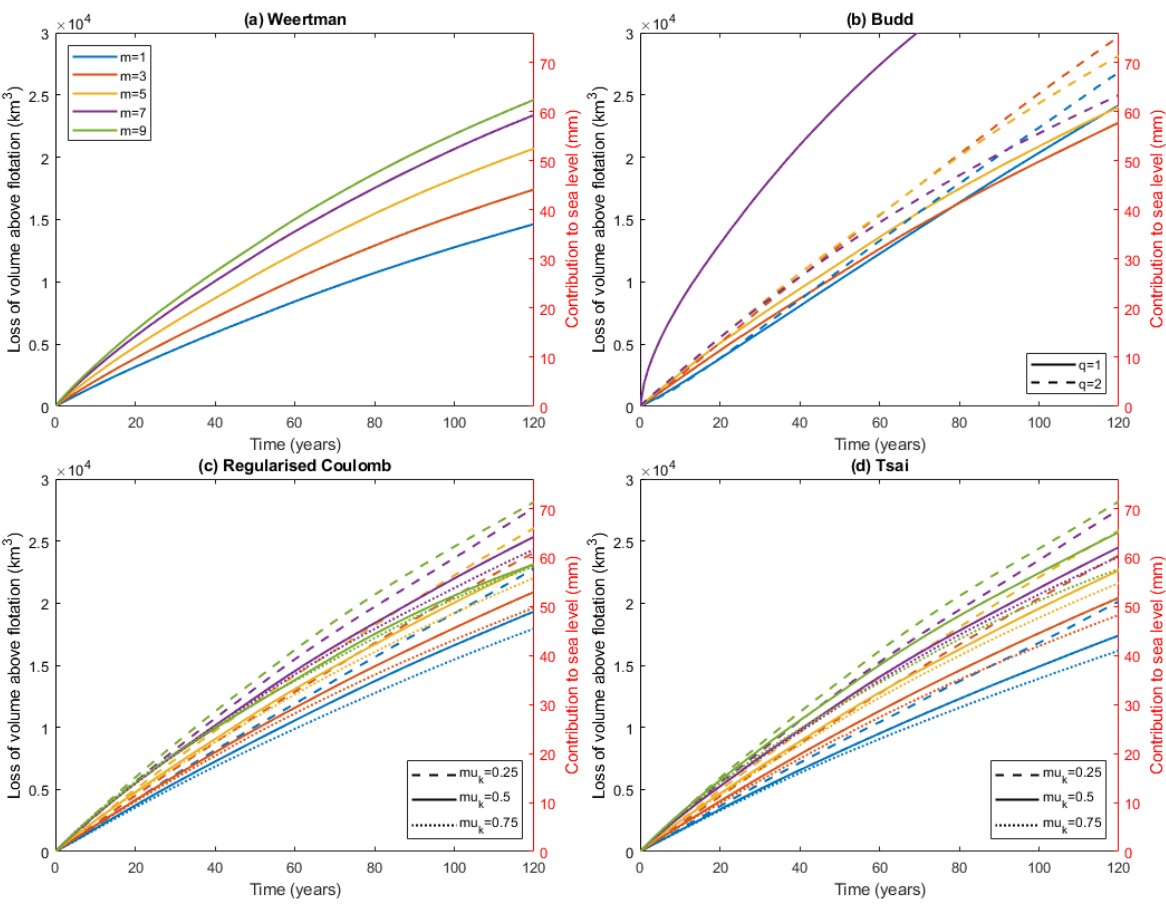

**Figure 2.** VAF loss in the *Control Branch* for every set of parameter choices in the sliding laws.





| Sliding law | Value of $m$ | | | | |
|---|---|---|---|---|---|
| | 1 | 3 | 5 | 7 | 9 |
| Weertman | 37.11 | 44.13 | 52.47 | 59.30 | 62.39 |
| Budd, $q = 1$ | 61.27 | 57.76 | 60.95 | 103.82 | - |
| Budd, $q = 2$ | 67.97 | 75.24 | 71.34 | 63.34 | - |
| Reg. Coulomb, $\mu_k = 0.25$ | 57.76 | 61.01 | 66.01 | 70.14 | 71.33 |
| Reg. Coulomb, $\mu_k = 0.5$ | 49.00 | 52.97 | 58.71 | 64.24 | 58.61* |
| Reg. Coulomb, $\mu_k = 0.75$ | 45.50 | 49.80 | 55.85 | 61.68 | 58.20* |
| Tsai, $\mu_k = 0.25$ | 51.06 | 60.41 | 65.56 | 69.74 | 71.42 |
| Tsai, $\mu_k = 0.5$ | 44.09 | 51.78 | 57.36 | 62.11 | 65.18 |
| Tsai, $\mu_k = 0.75$ | 41.02 | 48.29 | 54.79 | 60.17 | 57.68* |

**Table 1.** The VAF loss (mm sea level rise equivalent) at the end of the *Control Branch* simulations, for all experiments. Values marked with * indicate where different priors had to be used in initialisation.

of $m$ causing smaller VAF changes in all sliding laws other than Budd. The Regularised Coulomb and Tsai laws produce very similar curves, becoming more similar with higher values of $m$. When using the same values of $m$, both produce larger mass loss than the Weertman law even for the highest friction coefficient in our ensemble ($\mu_k = 0.75$), although the gap between the two becomes significantly smaller with higher values of $m$. There is also a systematic pattern of higher values of $\mu_k$ (i.e. higher

friction) causing less change in volume above flotation. In a few cases, the curves do not fit the pattern. This is because during initialisation, a slight change had to be made in order for the inversion process to run successfully in the $m = 9$ cases for the Regularised Coulomb law with $\mu_k = 0.5$ and $0.75$ and for the Tsai law with $\mu_k = 0.75$. These particular cases should perhaps be discounted when assessing patterns in the results, but are presented here to demonstrate a dependence on the initialisation.

The Budd sliding law shows less systematic behaviour with respect to the value of $m$, but a higher value of $q$ produces a larger change in VAF, for values of $m$ up to 5. When $m$ becomes too high, the Budd sliding law produces very high ice loss, as demonstrated when $m = 7$ and $q = 1$ in Figure 2. Raising $m$ further soon makes the sliding law unusable. For both values of $q$ in our ensemble, simulations using $m = 9$ ran into convergence issues early on, and could not be completed.

Turning our attention to the responses to changes in ice shelf forcing, the differences in VAF loss for the entire ensemble of experiments are displayed in Figure 3 and Table 2. The results are presented with reference to the *Control Branch* in each case. Since the *Control Branch* is run for 20 years before any changes are made to the forcing, the system state at the point of pertur­bation will be slightly different in each case. It is therefore useful to look at VAF differences in terms of a percentage change compared to the *Control Branch*, rather than as a volume in $\mathrm{km}^2$. The Weertman sliding law displays the smallest response


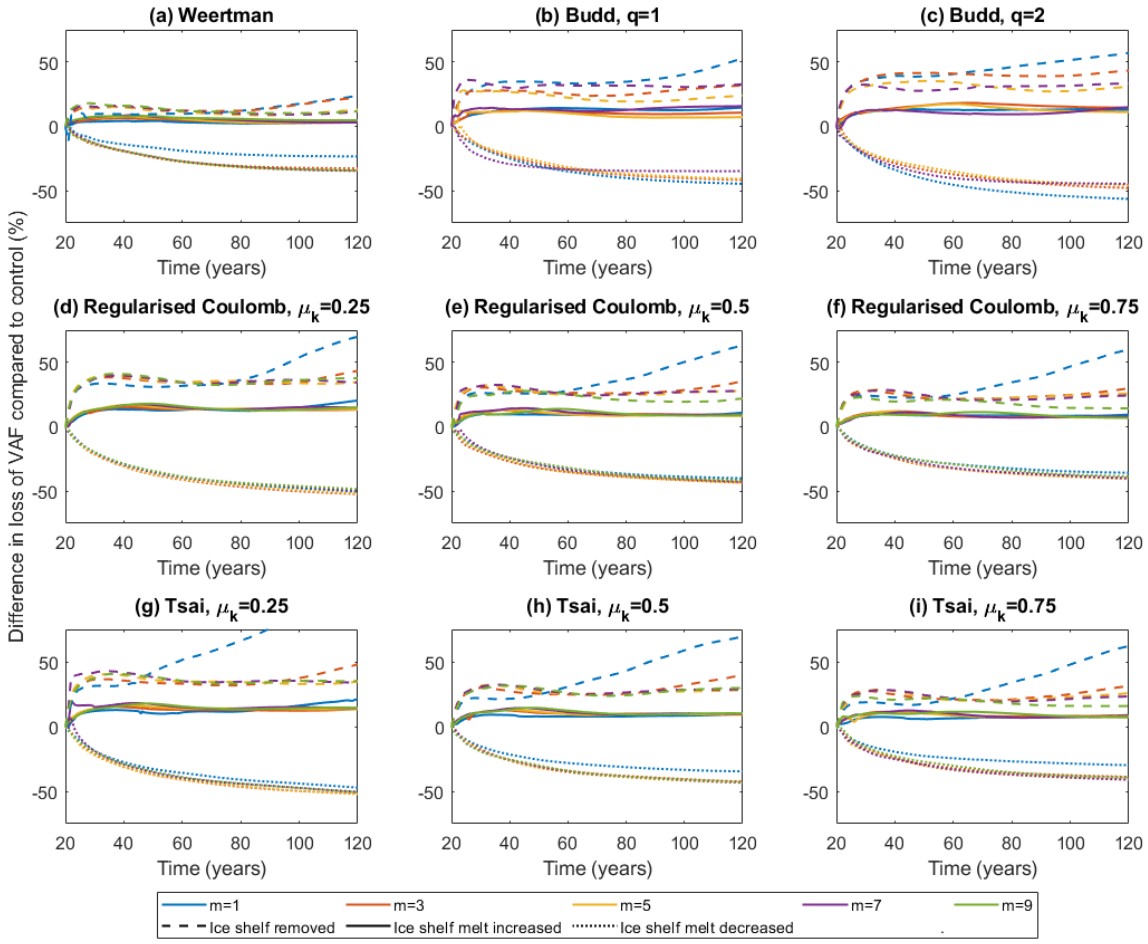

**Figure 3.** Additional VAF loss compared to the *Control Branch* after the changes in forcing are introduced, for the entire ensemble of sliding laws and parameter choices.

to all changes, while at a glance the other three sliding laws appear to respond quite similarly, when using the same values of $m$.

For the *Decreased Melt Branch*, it is notable in most panels of Figure 3 that a larger difference is seen between linear sliding ($m = 1$) and the non-linear versions ($m > 1$), than between any two non-linear cases, which all respond very similarly. For the Weertman law, the VAF loss difference at the end of the simulation compared to the *Control Branch* is -23.35 % for $m = 1$ and between -32.67 % and -34.52 % for the other values of $m$. For the Tsai and regularised Coulomb laws, this gap between the $m = 1$ response and other values of $m$ is larger for higher values of $\mu_k$, but the difference from the *Control Branch* is larger for lower values of $\mu_k$. The difference in response with varying friction coefficient is not linear. As an example, when $m = 3$, the VAF loss difference for the Regularised Coulomb law is -52.32 % with $\mu_k = 0.25$, -43.36 % for $\mu_k = 0.5$ and -39.88 % for



| Sliding law | Decreased Melt Branch | | | | | Increased Melt Branch | | | | | Ice Shelf Removed Branch | | | | |
|---|---|---|---|---|---|---|---|---|---|---|---|---|---|---|---|
| Value of $m$ | 1 | 3 | 5 | 7 | 9 | 1 | 3 | 5 | 7 | 9 | 1 | 3 | 5 | 7 | 9 |
| Weertman | -23.35 | -32.68 | -34.07 | -34.52 | -34.39 | 3.28 | 3.72 | 3.10 | 3.05 | 4.82 | 23.70 | 22.33 | 12.90 | 11.42 | 11.79 |
| Budd, $q=1$ | -44.58 | -41.61 | -40.77 | -34.70 | - | 14.66 | 10.74 | 7.18 | 15.69 | - | 52.86 | 31.72 | 23.97 | 32.74 | - |
| Budd, $q=2$ | -56.32 | -47.89 | -45.59 | -44.57 | - | 13.14 | 14.65 | 11.11 | 14.90 | - | 56.98 | 43.30 | 30.87 | 33.40 | - |
| Reg. Coulomb, $\mu_k=0.25$ | -50.33 | -52.32 | -51.77 | -49.65 | -48.53 | 20.40 | 13.35 | 13.46 | 14.87 | 14.96 | 69.95 | 43.44 | 34.75 | 34.26 | 37.68 |
| Reg. Coulomb, $\mu_k=0.5$ | -40.00 | -43.36 | -43.30 | -42.62 | -41.71* | 10.98 | 9.33 | 8.39 | 8.98 | 8.89* | 63.20 | 35.38 | 28.00 | 27.88 | 21.96* |
| Reg. Coulomb, $\mu_k=0.75$ | -35.69 | -39.88 | -40.53 | -39.98 | -38.44* | 9.37 | 7.66 | 7.37 | 7.70 | 6.85* | 59.89 | 29.75 | 25.80 | 24.21 | 14.33* |
| Tsai, $\mu_k=0.25$ | -47.12 | -51.820 | -52.13 | -50.35 | -50.71 | 21.15 | 13.50 | 13.35 | 14.20 | 14.54 | 103.01 | 48.04 | 34.80 | 33.88 | 35.27 |
| Tsai, $\mu_k=0.5$ | -34.57 | -42.43 | -43.45 | -43.26 | -42.93 | 9.58 | 9.75 | 9.17 | 9.74 | 10.25 | 69.53 | 39.97 | 28.19 | 28.67 | 29.75 |
| Tsai, $\mu_k=0.75$ | -29.87 | -38.87 | -40.93 | -41.01 | -39.25* | 8.85 | 8.26 | 7.56 | 7.38 | 7.09* | 62.10 | 31.34 | 25.81 | 23.26 | 15.89* |

**Table 2.** The percentage differences in VAF at the end of simulation compared to the *Control Branch*, for all experiments. Values marked with * indicate where different priors had to be used in initialisation.

$\mu_k = 0.75$. Taking $\mu_k = 0.5$ as a midpoint, lowering the friction causes a significantly larger difference in the response than raising it. For the Budd law, there is more variability and a slightly larger percentage difference with $q = 2$ compared to $q = 1$. The effect of $m$ is also reversed compared to the other laws, with $m = 1$ responding more than higher values.

The *Increased Melt Branch*, as the less extreme of the three forcing changes, generally causes the lowest differences in the responses. The VAF loss differences using the Weertman law range from 3.05 % to 4.82 %. The obvious outliers are in the Regularised Coulomb and Tsai laws with low friction ($\mu_k = 0.25$), where the simulations using $m = 1$ start to display an increased response towards the end of the run, and the Budd law for which there is more variability in the responses for different values of $m$, particularly when $q = 1$. Using Regularised Coulomb with $m = 3$ as an example again, the response changes with $\mu_k$ in a similar way to that seen in the *Decreased Melt Branch*. For $\mu_k = 0.25$, the VAF loss is 13.34 % greater than in the *Control Branch*, and increasing the friction to $\mu_k = 0.5$ and $\mu_k = 0.75$ decreases the difference to 9.33 % and 7.66 %, respectively.

The largest variability is found in the *Ice Shelf Removed Branch*. The Budd sliding law shows similar behaviour here as in the other two branches, with lower values of $m$ and higher values of $q$ causing larger responses. The other sliding laws all display a clearly new behaviour. In all cases, the general trend seen in other branches holds initially, with higher values of $m$ causing greater changes to the VAF. However, the response for lower values of $m$ in this branch becomes much larger at some point during each simulation. This point is reached earlier with lower values of $\mu_k$ in the Tsai and Regularised Coulomb laws, and earliest when using the Tsai law, which produces just over twice the VAF loss of the *Control Branch* when $m = 1$. Within these two laws, the difference is far more evident with $m = 1$. Using the Weertman law, the difference in response is lower overall and very similar for $m = 1$ and $m = 3$, which have VAF loss 23.70 % and 22.33 % higher than the *Control Branch*,

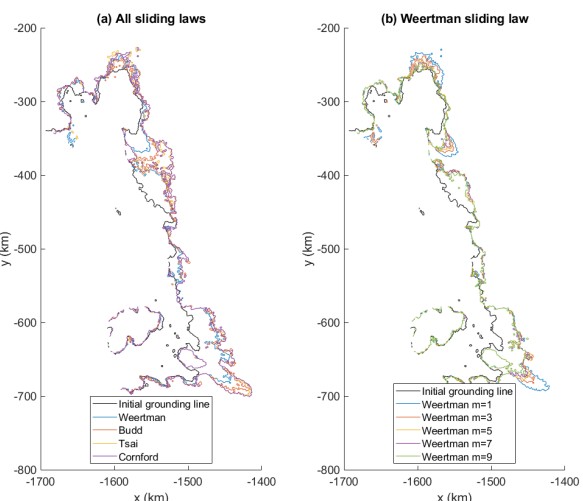

**Figure 4.** Grounding lines at 120 years in the *Ice Shelf Removed Branch* for (a) all sliding laws using $m = 3$, $q = 1$ and $\mu_k = 0.5$ and (b) different values of $m$ in the Weertman sliding law. The initial grounding line (t=0) is also indicated.

respectively, while the other values of $m$ are in the range 11.42-12.90 %. The effects of altering $\mu_k$ are similar to those in the other branches. with the difference from the *Control Branch* higher with $\mu_k = 0.25$ than when $\mu_k = 0.5$, and the difference when $\mu_k = 0.75$ a little lower.

The VAF values are just one metric by which one can assess the system. Another option is to look at the positions of the grounding lines at the end of simulations. Figure 4 shows the grounding lines for all sliding laws using the values $m = 3$, $q = 1$ and $\mu_k = 0.5$, and for all values of $m$ within the Weertman sliding law, in the *Ice Shelf Removed Branch*. Since this shows the response to the most extreme forcing change, this is the case for which differences are most evident. When comparing the sliding laws using the same parameter values, it is clear that the Weertman law causes the least retreat, and that the Tsai and Regularised Coulomb laws cause the most, ending up in very similar positions. For the most part, the grounding line produced by the Budd law lies somewhere between the two. When looking at the grounding lines for the Weertman law using different parameters, it can be seen that the grounding line retreats less far for higher values of $m$. This trend is much more noticeable around the Pine Island and Kohler Glaciers, while the grounding lines are located in very similar positions around Thwaites Glacier and the Crosson Ice Shelf.

We can also look directly at model outputs of ice thickness and velocities. Figure 5 displays the differences in thickness and speed at the end of the *Ice Shelf Removed Branch* for each non-linear version of the Weertman law, compared to that of the linear case ($m = 1$). Across much of the domain, the ice is thinner and moves faster when higher values of $m$ are used in the sliding law. However in some areas, specifically Pine Island and Kohler Glaciers, the opposite is true. The differences between

**Figure 5.** Differences in final ice thickness and speed in the *Ice Shelf Removed Branch*, for the Weertman sliding law. All other values of $m$ are compared to the case where $m = 1$. Areas which are not grounded in either case are not shaded as part of each comparison.

experiments using successive values of $m$ become less as $m$ increases, so the largest difference is between the linear case and $m = 3$, while the difference between the $m = 7$ and $m = 9$ cases is much smaller.

# 4 Discussion

## 4.1 The whole domain

As outlined in the previous section, the results display fairly systematic behaviour in many respects. Some of the patterns are simple, such as higher friction coefficients $\mu_k$ resulting in smaller changes to the system. Other trends are more complicated. In particular, there is an apparent inconsistency in the effects of the value of $m$. When the melt is decreased, it is clear that

higher values of $m$ respond more to the change, with the system gaining more ice when $m > 1$. However, with an increased



melt rate, and especially with the removal of the entire ice shelf, low values of $m$ cause a larger response.

The Budd sliding law responds in ways which set it apart as an outlier among those tested. This is most likely because it re­lies on the value of effective pressure, $N$, throughout the domain. In our model $N$ is calculated assuming a perfect hydrological

connection with the ocean, so varies proportionally with ice thickness when grounded. This is a reasonable assumption close to the grounding line, but can become unrealistic further upstream. With more accurate information about the effective pressure at the base of the ice sheet, the Budd law would likely produce results more consistent with the other sliding laws. However, in our case the assumption causes erratic behaviour, and the Budd law is not considered in detail during our discussion. The Tsai and Regularised Coulomb sliding laws also include $N$, but in these laws the Coulomb component of the equation is only

relevant in close proximity to the grounding line, with a transition into Weertman-style sliding upstream.

Excluding the Budd sliding law, all of the *Control Branch* experiments (Figure 2) show a similar evolution in the volume of ice contained within the domain. After 120 years of transient simulation, the loss of volume above flotation ranges from about $37\,\mathrm{mm}$ to $71\,\mathrm{mm}$ sea level rise equivalent, with a mean value of $57.05\,\mathrm{mm}$. The standard deivation is $8.68\,\mathrm{mm}$, giving a coef­

ficient of variation 0.15 which implies moderate variation in rates of change in the system. But no matter which sliding law or parameters are used, the direction of evolution is always the same, suggesting that the choice of sliding parameterisation does not introduce a level of uncertainty which would affect predictions of the broader patterns of future ice movement in a system forced by ice shelf melt rates in a realistic range. Indeed, it has been shown by Hill et al. (2021) that far larger uncertainties are generally introduced by oceanic forcing than by parameters of the ice flow dynamics.


The difference in rates of change when varying basal sliding conditions can be seen more clearly when perturbations are introduced. Again, all sliding laws react in broadly the same way, but with different rates and magnitudes of change. From this, we can say that the choice of sliding law does not impact predictions of how an ice sheet system will change in terms of the direction of evolution, but it does affect how fast the changes occur. It is also worth noting that the large differences in

behaviour for low values of $m$ in the *Ice Shelf Removed Branch* are due to an unrealistic extreme forcing accelerating the rate of change in the system, and that the results in the more moderate *Increased Melt Branch* do not differ so much.

The significantly larger percentage rises resulting from lowering the buttressing force of the ice shelves when using the lower values of $m$ cannot be accounted for by differences in the state of the system at the point of perturbation. These are

differences between the rates at which the forward simulations respond after the perturbations are introduced. It is likely that this is driven by the large speed reductions of Pine Island as the value of $m$ increases. The grounding line positions (Figure 4) and differences in ice thickness and speed (Figure 5) clearly suggest that the modelled response to different $m$ values, and the sensitivity to basal sliding representation in general, is affected by regional differences.





## 4.2 Regional Variation

To investigate differences within the ASE, we divide the domain into four regions, as indicated by the colours on Figure 1. These regions are the catchment basins of three glaciers - Pine Island, Thwaites and Kohler - and the combined catchments of the small glaciers which flow out to the Crosson Ice Shelf. Figure 6 shows the magnitude of differences in VAF for these four regions. Pine Island Glacier has the largest overall ice loss, and thus contributes heavily to the behaviour seen for the whole domain in the previous section.


Figure 7 shows the percentage differences in VAF loss compared to the *Control Branch* for all other branches using the Weertman sliding law. Figure 8 displays the same information in the case of using the Tsai sliding law with $\mu_k = 0.5$. The results from the Regularised Coulomb law are very similar to those from the Tsai law. We can see a variety of responses across the domain, showing that regional factors can play an important role in how sliding laws react to changes. In terms of percent-

age increases, Kohler Glacier responds the most by far to changes involving a loss of buttressing, with up to 150 % greater

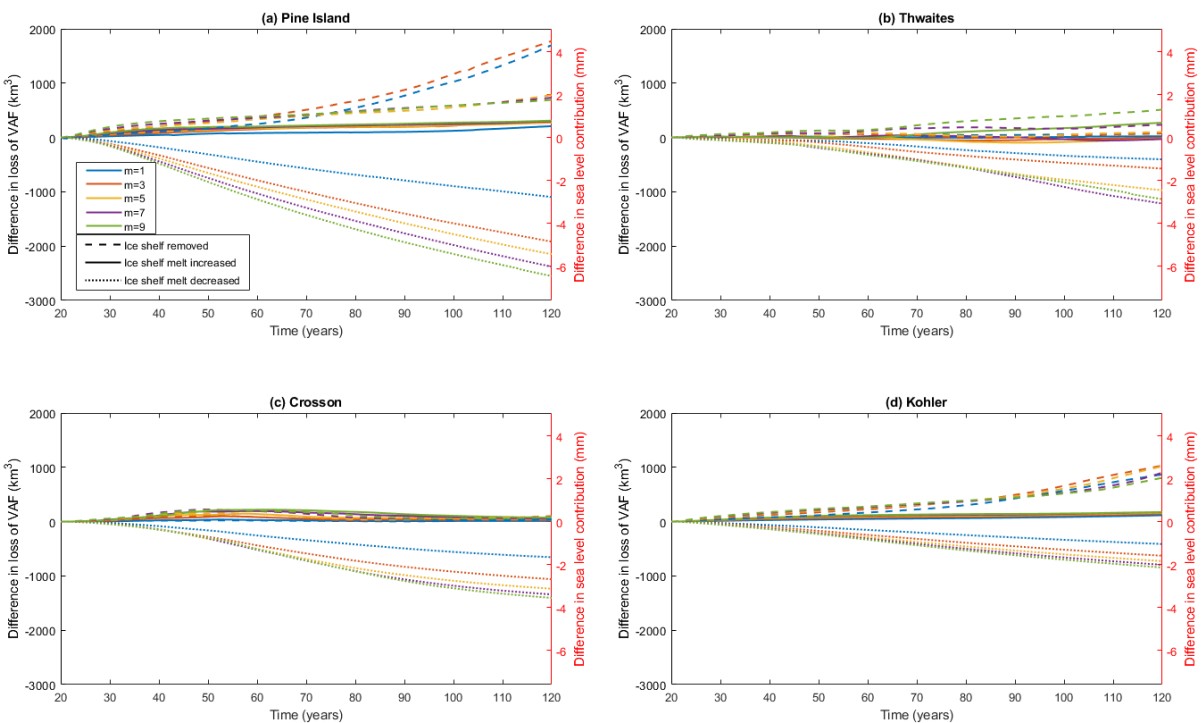

**Figure 6.** Additional VAF loss compared to the *Control Branch* for the Weertman sliding law with different values of $m$. The contributions of each region within the ASE domain are shown.



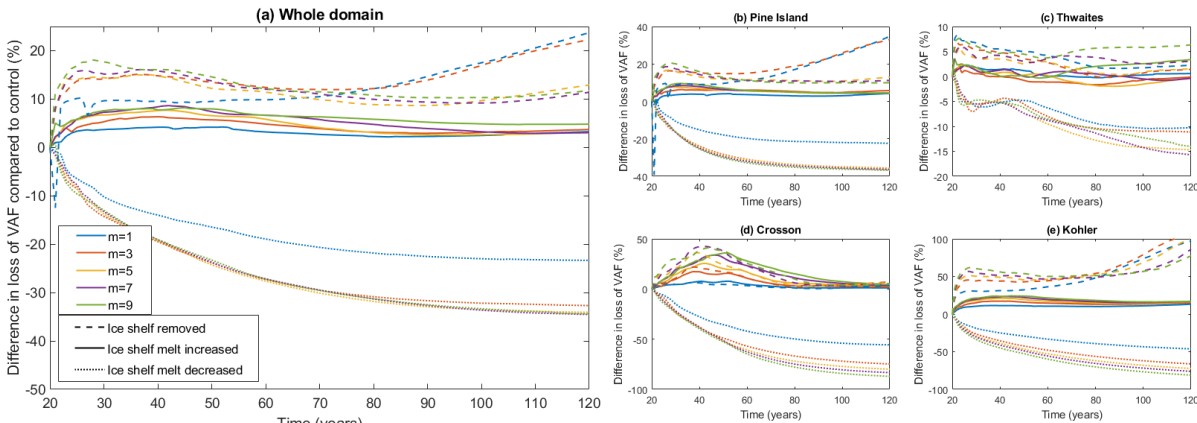

**Figure 7.** Additional VAF loss compared to the *Control Branch* for the Weertman sliding law with different values of $m$. The contributions of each region within the ASE domain are shown. Note that the $y$-axes for each region use different scales.

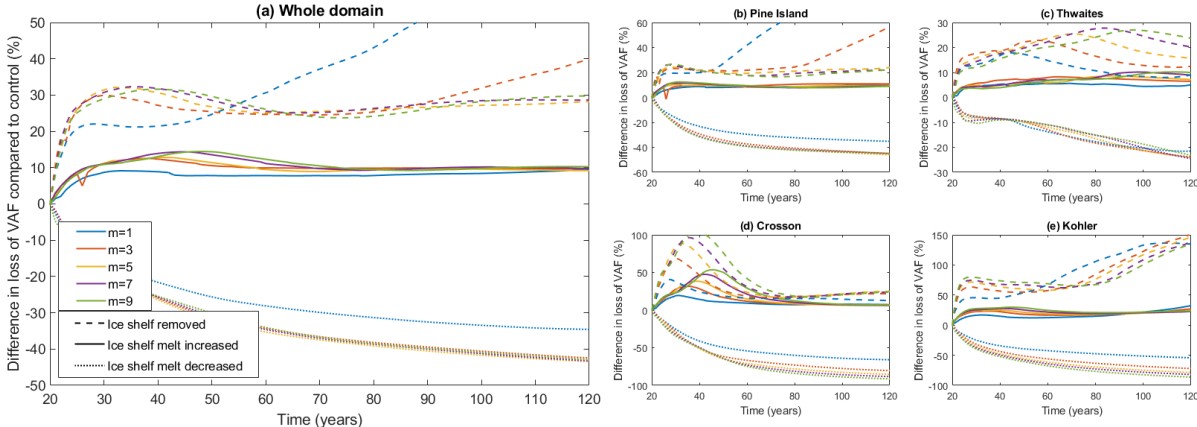

**Figure 8.** Additional VAF loss compared to the *Control Branch* for the Tsai sliding law with $\mu_k = 0.5$ and different values of $m$. The contributions of each region within the ASE domain are shown. Note that the $y$-axes for each region use different scales.

VAF using the Tsai law. The differences in VAF for Thwaites and Crosson at the end of the simulations is very small, less than 10 % for the Weertman law, and 30 % for the Tsai law.

The large changes when the ice shelf is removed and $m$ has a low value are not universal. They occur in the Pine Island and Kohler Glaciers, the same locations in which the largest grounding line retreat is seen in Figure 4. Thwaites Glacier, and those flowing into the Crosson Ice Shelf, do not display the same behaviour. Looking at flowlines through Pine Island Glacier, we discover that 50 years into the simulations, the glacier is grounded on a ridge in the bathymetry. With low values of $m$, the ice unpins from this point earlier, producing the sudden rise in VAF differences. The region-specific effect at play





here could be the bathymetry itself. Nias et al. (2018) found that for Pine Island Glacier, using different bathymetry datasets

can effect whether more or less ice is lost with increasing values of $m$. They conclude that the buttressing force of a bump in one bathymetry dataset is amplified by larger values of $m$, slowing down the flow. Our bathymetry is a different product to either tested in that study, but it is likely that there is a similar cause for the pattern observed in our results and that such bathymetric features do not exist around Thwaites Glacier and Crosson Ice Shelf, so the same effect is not seen in those regions.

The relationship between the value of $m$ and the rates of retreat and ice loss is an issue which could be explored further. Nias et al. (2018) identifies local bathymetry as a likely source of differences in behaviour, but there could be other factors to consider, such as variation in the basal sliding parameter $C$ or flow rate factor $A$. We note that even a small change required for successful inversion in three of our cases caused the responses in those cases to fall out of line with the patterns seen in the rest, suggesting a high sensitivity to changes in the inverted fields. Whatever the cause, we can confidently say that the relation-

ship contains complexities and can be affected by regional differences, so there is no simple statement to be made about what the general effect of increasing $m$ will be on a system. We believe the regional variance in the effect of $m$ supports the idea that $m$ should perhaps be treated not as a universal constant, but as a distributed field of values, as suggested by De Rydt et al. (2021). At the very least, different glaciers may need to use different values in order to reproduce their observed rates of change.

In general, whatever differences are seen when increasing the value of $m$, they are smaller between higher values. This can be seen in the control experiments (Figure 2), in which the $m = 7$ and $m = 9$ cases are closer together than those of lower sequential values in our ensemble. It is also seen clearly in most panels of Figure 3 for the *Decreased Melt Branch* that the $m = 1$ case is more different from the other cases, and in the differences in final speed and thickness presented in Figure 5. This demonstrates that when increasing the value of $m$ in sliding laws, the response converges. With large values of $m$, the

Weertman law is often said to approach the behaviour of a Coulomb friction law in this way.

An observation which appears strange at first is that when a change in forcing increases the melt rate or removes the buttressing ice shelf entirely, higher values of $m$ can cause larger VAF losses but at the same time less retreat of the grounding line. We see this around Thwaites glacier when using the Weertman sliding law, for which the $m = 9$ experiment produces the largest

VAF loss in the *Control Branch* (Figure 2a), the largest percentage response in the *Ice Shelf Removed Branch* (Figure 7c), and the greatest ice speed and thinning (Figure 5), yet the least grounding line retreat (Figure 4b). We suggest that when more ice is lost from upstream areas, it crosses the grounding line at a rate faster than the ice shelf is melting away, thus causing a buildup of ice in this region. Because the ice around the grounding line is thicker, the glaciers stay grounded. In this way, it is possible for higher values of $m$ to cause glaciers to lose a greater volume of ice while at the same time retaining a larger grounded area.


In most cases, we see that the responses in each region follow the same patterns using the Weertman and Tsai laws. There are some differences, most notably in the response of Thwaites, for which a significantly different pattern can be seen. However, this is another effect of the greater speed of response using the Tsai law. As can be seen in the grounding lines of Figure 4,





for the Tsai and Regularised Coulomb laws, the ice retreats far enough for the Thwaites and Pine Island ice shelves to merge. The two glaciers become much more closely linked, and this has an effect on the evolution of Thwaites. Even before the ice sheets merge, the retreat of the small western section of Pine Island starts cutting into the Thwaites region, as we have defined the regions by present-day boundaries. The same behaviour would be seen in the Weertman case if the model was run for longer. This is a significant example of the importance of differing rates of change caused by the choice of basal sliding parameterisation.

## 5 Conclusions

Parameter choices in sliding laws have an important effect on the speed and magnitude of responses to changes in the forcing of the ice system, although the exact nature of the effect can be highly region-dependent. The Weertman, Tsai and Regularised Coulomb laws all display similar behaviours in the effects of changing the values of $m$ and $\mu_k$. The Budd sliding law responds differently to the other laws, due to the simplistic hydrological condition used in our model, but in general raising the value of $q$ increases the magnitude of the response.

Changing the value of the friction coefficient $\mu_k$ produces a very structured response in all cases. Simulations with lower values of $\mu_k$ respond more to changes in forcing. The relationship is not linear, with the differences in response compared to the *Control Branch* converging as $\mu_k$ increases. This is to be expected, as there will be a convergence to the limit at which friction is sufficiently high to allow no basal sliding at all.

Changing the value of the sliding exponent $m$ produces a more varied range of responses, and is affected by regional differences in bathymetry as well as the inverted fields $A$ and $C$. In the case of lowering the ice shelf melt rate and allowing increased buttressing, $m = 1$ is a notable outlier, producing less difference in VAF compared to the *Control Branch*, while higher values of $m$ respond very similarly to the change. The response to increased melting or removal of the ice shelf also systematically shows higher values of $m$ responding more in terms of VAF loss, up to a point at which the trend switches. This is due to regional changes at Pine Island and Kohler Glaciers, where lower values of $m$ cause more rapid retreat along the main glacier trunks. Thwaites Glacier, and those flowing out to the Crosson Ice Shelf, do not display this behaviour.

Despite the differences in our results when using different parameterisations of basal sliding, with the highest VAF loss at the end of the *Control Branch* being almost twice that of the lowest, there is also a degree of similarity between them. In broad terms, the system always reacts in the same way to a particular change in forcing, no matter which sliding law or parameter values are used. The magnitude and speed of these reactions vary, but the response is clearly bounded. Our results suggest that ice sheet models, if initialized by inverting for sliding-law parameters using present-day geometry and surface velocities and driven by forcing in a realistic range, follow approximately the same trajectory for at least several decades and up to a century, irrespective of the choice of sliding law. In this sense, the predictions of ice sheet models about near-future ice loss are



reasonably well constrained despite our incomplete knowledge of basal sliding processes. The implication is that initialized ice sheet models therefore have predictive power over shorter times scales of several decades. Further work is required to determine if this holds when other poorly constrained processes such as calving and ice damage are included.

*Code availability.* Version 2019b of the source code for Úa is available at http://doi.org/10.5281/zenodo.3706623 (Gudmundsson, 2020).

*Author contributions.* JMB ran the model simulations and led the writing. Both authors were involved in devising the experiments and editing the manuscript.

*Competing interests.* The authors declare no competing interests.

*Acknowledgements.* This work is from the PROPHET project, a component of the International Thwaites Glacier Collaboration (ITGC).
Support from Natural Environment Research Council (NERC: Grant NE/S006745/1). This is ITGC Contribution No. 074.



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
