# Peer review of "The predictive power of ice sheet models and the regional sensitivity of ice loss to basal sliding parameterisations: A case study of Pine Island and Thwaites Glaciers, West Antarctica"

_The Cryosphere, 2022_

## Referee Comment (RC2)

**Referee Comment on 'The predictive power of ice sheet models and the regional sensitivity of ice loss to basal sliding parameterisations: A case study of Pine Island and Thwaites Glaciers, West Antarctica'**

Doug Brinkerhoff

July 2022

**Summary**

In this manuscript, the authors present the results of sensitivity study in which they vary the choice of sliding parameterization along with the numerical values of the associated parameters and evaluate the different solutions that result for a pair of important glaciers in the Amundsen Sea embayment under several idealized geometric perturbations. They find that there are broad similarities in the trajectories of models using these different parameterizations, with the overall sign remaining the same for all cases, but with some qualitative differences as well as some regional peculiarities.

I found the paper to address an important question in ice sheet modelling and to be reasonably easy to follow. The results are interesting and provide some degree of comfort that, even though as modellers we are generally ignorant of the appropriate way to model sliding, there is some hope in West Antarctica that the models are still producing useful results. I have a handful of specific comments and concerns - some more serious than others - which I have included below.

**Comments**

**L21** Define effective pressure here rather than on line 33.

**L58** The initialism 'ASE' is first defined on line 88, below where it appears here.

**L65** It doesn't really make sense to say that a higher $m$ produces higher velocities unless it's qualified by a description of what's going on with the friction coefficient, which is, in this work and others, treated as an unconstrained spatially-varying free parameter. Can this notion of velocity increasing with $m$ be made a bit more rigorous?

**L97–99** There are a few too many commas in this sentence.

**Eqs. 1–3** These equations are defined again later, albeit in slightly different form, which is a bit confusing. I understand why they are here for exposition, but maybe they could either be described qualitatively, or not numbered, or something else to make clear that they are not necessarily the equations actually implemented.

**L103** While most readers of this paper will likely already understand the assumptions of the SSA, for completeness they should be briefly mentioned here.

**Eq. 4** Probably not necessary to include this, unless you are going to state the full set of governing equations. I think it's okay just having it in the Ua reference.

**L108** What flux and stress conditions are imposed at the inland boundary?

**L109** It is not very clearly described how calving is treated in this model. Is the calving front held fixed or evolved? How?

**L112** We're not given any context into how these densities are used (and are not mentioned again in the text). These should either be omitted or the full model needs to be described.

**Eqs. 6–10** It's not clear to me what the practical difference is between $C$ and $\mu$. While they may have different physical interpretations, both would appear to be friction coefficients. However, in this paper, one is treated as a spatially varying field to be inferred from velocity observations while the other is a scalar parameter. It would be helpful to try to clarify this in the text. This also implies that different sliding laws have substantially different degrees of flexibility and, as a result presumably exhibit different degrees of fidelity to observed surface velocities. It would be worthwhile to comment on this issue as well: are the differences in evolution really the result of the different sliding laws, or is it because some laws match observed velocities better than others?

**Eq. 7–10** I think that the redefining of $\beta^2$ is a confusing choice of notation, given that $\beta^2$ usually means what you call $C$. Furthermore, this substitution makes the dependence of the various laws on the sliding exponent $m$ much less transparent. I suggest writing the laws in a way that makes this dependence more plain (perhaps by just substituting the given expression for $\beta^2$ back into the equations).

**L130–131** As written, the Budd law only becomes similar to the Coulomb law for high $m$ when $q = m$, which these experiments don't explore.

**Eq. 12** The choices made for $p_{k,prior}$ need to be shown here. Is it a spatially varying field or is it a scalar? how is it selected? Furthermore, how well the resulting velocity solutions correspond to observations (and to one another between sliding laws) needs to be quantified here.

**L143** Total area of what? Perhaps introduce a symbol ($\Omega$ is frequently used) to represent the model domain.

**L148** I don't know what it means for the system to 'settle', nor is it clear what it means for a strongly non-linear system as this to 'evolve steadily'. The manuscript needs to include a more detailed description of the criteria used for assessing whether the so-called 'transients' associated with data-model incompatibility in time-dependent simulations have reached a sufficiently small degree of influence so as to render the resulting simulations plausible.

**L152–157** I would suggest using the 'description' LaTeX environment (or equivalent) here rather than 'itemize'; the two different hyphens look weird together.

**L165** Median might be a better choice than mean here, since the sample contains some outliers (e.g. Budd $q = 1, m = 7$).

**L170–174** It would be better to be specific about what change was made to convince the solver to converge so that a reader can properly assess the impacts that such a change would have when compared to the other simulations.

**L186 and elsewhere** Perhaps two significant digits is sufficient when reporting percentage changes.

**Fig. 5** This may be a comment on Ua more than the present manuscript, but Fig. 5 shows clear indication of numerical artifacts in the thickness field (presumably the result of an unstable continuity discretization). If this is indeed the case, this needs to be mentioned in the text and the potential implications mentioned (which might be significant: after all, the flux across the grounding line depends quite sensitively on the thickness there).

**L245–246** It is not immediately clear why the oceanic pressure condition shouldn't apply away from the grounding line. In any case, a reference for this claim is needed.

**L254** typo: 'deivation'

**L316–324** It might be worth mentioning that this effect of faster transport of upstream ice to the grounding line is probably a strictly transient phenomenon.

**L358–362** The conclusion that the sliding law choice doesn't matter much over centennial time scales is reasonably well supported by the results of this paper, but this might not be true in non-WAIS scenarios; it's worth remembering that this is a system that is already undergoing a strong and non-linear transient response and may be much less sensitive to the types of choices explored here than other places. I think that the language of this section needs to be tempered a bit to reflect this.

---

## Author Comment (AC2)

**Response to reviewers**

We thank the reviewers for the time and effort taken to read through and comment upon this manuscript, and for their helpful comments which have helped us to make improvements. We attach a document showing changes made to the manuscript in response.

Reviewer 1

This paper present a suit of synthetic simulations of the Amundsen sea sector of the Antarctic Ice-Sheet. The objective is to study the sensitivity of centennial scale ice-sheet simulations to the basal friction law formulation.

The results show that the predicted volumes and rates of volume change are sensitive to the friction law, but also to the exponent *m,* with complex patterns at the regional scale. However they found that all simulations follow the same trajectories (i.e. mass loss or gain) for a given perturbation, giving some confidence that ice-flow models have some predictive skills at these time-scales.

The paper is well structured and written and the results are convincing. I have only few minor comments.

First, effective pressure dependent friction laws, by definition, depends on the basal water pressure, and thus this requires a specific model. However, it's only in the discussion section that we learn the assumption that has been made, i.e. perfect connectivity. This assumption should already be presented in the model set-up section.

A good point! This assumption is now mentioned in the model setup.

Second, for the *Decreased Melt Branch* experiments, I suspect that this perturbation might lead to an advance of the grounding lines ? However, the friction parameters are not constrained by observations in initially floating areas, so the differences between the simulations might more depends on the assumptions for the different priors than on the friction law or *m* ?

Decreasing the melt does indeed cause advance, but we do not believe this is a problem for our experiments. One advantage of allowing the model to run for 20 years before introducing the perturbations is that in most places the grounding line retreats during this time, and the initial response to the perturbation occurs at a point where the sliding parameter is constrained. A new paragraph has been added to the discussion to address this.

Minor comments :

- Page 8, line 171 : *« a slight change had to be made in order »*. Please explain here what are this slight chnages ; I think we read latter that this is changes in the prior ?

The changes to the prior are now explained as part of the overview of experiments.

- Page 13, last paragraph : I don't understand the meaning of the first two sentences.

The sentences have been edited to hopefully make the meaning clearer.

Reviewer 2

In this manuscript, the authors present the results of sensitivity study in which they vary the choice of sliding parameterization along with the numerical values of the associated parameters and evaluate the different solutions that result for a pair of important glaciers in the Amundsen Sea embayment under several idealized geometric perturbations. They find that there are broad similarities in the trajectories of models using these different parameterizations, with the overall sign remaining the same for all cases, but with some qualitative differences as well as some regional peculiarities.

I found the paper to address an important question in ice sheet modelling and to be reasonably easy to follow. The results are interesting and provide some degree of comfort that, even though as modellers we are generally ignorant of the appropriate way to model sliding, there is some hope in West Antarctica that the models are still producing useful results. I have a handful of specific comments and concerns - some more serious than others - which I have included below.

Comments:

L21 Define effective pressure here rather than on line 33.

The definition has been moved.

L58 The initialism 'ASE' is first defined on line 88, below where it appears here.

The text has been changed to avoid use of the acronym before definition.

L65 It doesn't really make sense to say that a higher m produces higher velocities unless it's qualified by a description of what's going on with the friction coefficient, which is, in this work and others, treated as an unconstrained spatially-varying free parameter. Can this notion of velocity increasing with m be made a bit more rigorous?

A sentence has been added to explain that the sliding parameter C is set using initialisation methods based on matching to observations, and a clarification made that the higher velocities referred to are in forward simulation, rather than the initial velocity field. Hopefully this makes the meaning clearer.

L97–99 There are a few too many commas in this sentence.

This has been re-written.

Eqs. 1–3 These equations are defined again later, albeit in slightly different form, which is a bit confusing. I understand why they are here for exposition, but maybe

they could either be described qualitatively, or not numbered, or something else to make clear that they are not necessarily the equations actually implemented.

The numbering has been removed from these equations, to distinguish them from those used in the model which are introduced later.

L103 While most readers of this paper will likely already understand the assumptions of the SSA, for completeness they should be briefly mentioned here.

An additional descriptor of "vertically-integrated" has been added to SSA in this sentence, and a reference added to MacAyeal(1989).

Eq. 4 Probably not necessary to include this, unless you are going to state the full set of governing equations. I think it's okay just having it in the Ua reference.

The equation has been removed.

L108 What flux and stress conditions are imposed at the inland boundary?
L109 It is not very clearly described how calving is treated in this model. Is the calving front held fixed or evolved? How?

Descriptions of the boundary conditions (inland and calving front) have been added to the model setup.

L112 We're not given any context into how these densities are used (and are not mentioned again in the text). These should either be omitted or the full model needs to be described.

Clarification has been made that these densities are consistent with the BedMachine geometry. We believe they must be included in the setup as important physical parameters of the system.

Eqs. 6–10 It's not clear to me what the practical difference is between C and μ. While they may have different physical interpretations, both would appear to be friction coefficients. However, in this paper, one is treated as a spatially varying field to be inferred from velocity observations while the other is a scalar parameter. It would be helpful to try to clarify this in the text. This also implies that different sliding laws have substantially different degrees of flexibility and, as a result presumably exhibit different degrees of fidelity to observed surface velocities. It would be worthwhile to comment on this issue as well: are the differences in evolution really the result of the different sliding laws, or is it because some laws match observed velocities better than others?

We believe that calling C a "friction parameter" (as often the case in the literature) is misleading. We refer to C as a sliding parameter, or sometimes basal slipperiness. The C distribution incorporates behaviour due to the thermal and mechanical properties of ice as well as bed roughness. Since all of our sliding laws include the sliding parameter C, which is now explicitly noted in the text, the degrees of flexibility are comparable, whereas using a spatially varying friction coefficient would indeed add extra degrees of freedom to some laws (in particular the regularised Coulomb law) but

not others. The approach we use is common among modelling studies. There is not a large variation in how well velocities are matched in the inversion process by different sliding laws, and this is now mentioned in the model setup section.

Eq. 7–10 I think that the redefining of β2 is a confusing choice of notation, given that β2 usually means what you call C. Furthermore, this substitution makes the dependence of the various laws on the sliding exponent m much less transparent. I suggest writing the laws in a way that makes this dependence more plain (perhaps by just substituting the given expression for β2 back into the equations).

The description of the sliding laws has been simplified by putting β2 back into the equations as suggested.

L130–131 As written, the Budd law only becomes similar to the Coulomb law for high m when q = m, which these experiments don't explore.

The statement was referring to the Weertman law, which was not sufficiently clear. This has been rectified.

Eq. 12 The choices made for pk,prior need to be shown here. Is it a spatially varying field or is it a scalar? how is it selected? Furthermore, how well the resulting velocity solutions correspond to observations (and to one another between sliding laws) needs to be quantified here.

More detail about the priors has been added to the model setup section.

L143 Total area of what? Perhaps introduce a symbol (Ω is frequently used) to represent the model domain.

The total area of the domain. This has been clarified in the text.

L148 I don't know what it means for the system to 'settle', nor is it clear what it means for a strongly non-linear system as this to 'evolve steadily'. The manuscript needs to include a more detailed description of the criteria used for assessing whether the so-called 'transients' associated with data-model incompatibility in time-dependent simulations have reached a sufficiently small degree of influence so as to render the resulting simulations plausible.

This part has been rewritten. From experience across the userbase of the model we know that the initial transients in Úa are only present for the first year or so of a run, and our 20 year relaxation period allows plenty of time for these to become irrelevant.

L152–157 I would suggest using the 'description' LaTeX environment (or equivalent) here rather than 'itemize'; the two different hyphens look weird together.

A good suggestion. This has been done.

L165 Median might be a better choice than mean here, since the sample contains some outliers (e.g. Budd q = 1, m = 7).

A good point. The median value has been added, although in this case it was not much different from the mean.

L170–174 It would be better to be specific about what change was made to convince the solver to converge so that a reader can properly assess the impacts that such a change would have when compared to the other simulations.

The details of this have been added to the model setup section.

L186 and elsewhere Perhaps two significant digits is sufficient when reporting percentage changes.

The reported values in the text have been reduced to two decimal places.

Fig. 5 This may be a comment on Ua more than the present manuscript, but Fig. 5 shows clear indication of numerical artifacts in the thickness field (presumably the result of an unstable continuity discretization). If this is indeed the case, this needs to be mentioned in the text and the potential implications mentioned (which might be significant: after all, the flux across the grounding line depends quite sensitively on the thickness there).

There is indeed evidence of numerical artifacts in the thickness field upstream where the resolution is coarse. When compared to a finer mesh, this choice of resolution did not change our findings regarding the patterns of behaviour due to sliding law choices, and so does not affect the relevant outcomes within the context of this paper. However, it is important to note that the numerical artifacts are among the reasons that the experiments in this study should not be treated as predictive, something which has been made clear. This has been addressed both in the model setup and the discussion sections.

L245–246 It is not immediately clear why the oceanic pressure condition shouldn't apply away from the grounding line. In any case, a reference for this claim is needed.

The point has been expanded slightly, and a reference added.

L254 typo: 'deivation'

Corrected.

L316–324 It might be worth mentioning that this effect of faster transport of upstream ice to the grounding line is probably a strictly transient phenomenon.

It is unclear to us what is meant by this, so no change has been made to this paragraph.

L358–362 The conclusion that the sliding law choice doesn't matter much over centennial time scales is reasonably well supported by the results of this paper, but this might not be true in non-WAIS scenarios; it's worth remembering that this is a system that is already undergoing a strong and non-linear transient response and may

be much less sensitive to the types of choices explored here than other places. I think that the language of this section needs to be tempered a bit to reflect this.

A fair point. This conclusion has been qualified to recognise that it may not be true everywhere.

**Compare Results**

| Old File: | | New File: |
|---|---|---|
| **Paper_SlidingLaws.pdf** | versus | **PIG_Thwaites_Sliding_Laws.pdf** |
| **21 pages (2.07 MB)** | | **22 pages (2.54 MB)** |
| 04/08/2022 16:03:50 | | 05/09/2022 11:01:11 |

**Total Changes**

**328**

**Content**

102   Replacements

112   Insertions

89   Deletions

**Styling and Annotations**

4   Styling

21   Annotations

Go to First Change (page 1)

[revised manuscript text omitted]

---

## Author Response (AR2)

To the authors,

You have done a good job addressing the comments of the reviewers and I find the manuscript to be near ready for publication. I only provide some minor specific suggestions below.

Best regards,
Alex

Thank you for your suggestions. We have incorporated all of them in the latest version, as you will see in the accompanying track changes file.

Specific comments:

L12: all our => all of our

Figure 1: Consider using a color-blind friendly palette, or at least one with more subdued colors.

The previous version did appear to be colour-blind friendly when tested, but the subdued colours look a lot better either way! Thank you for this suggestion, which has greatly improved figure 1.

Eqs. 3 & 4: As a minor recommendation, consider reformatting to make Eqs. 2, 3 & 4 easily comparable. This could be done by giving all three equations a format like: $\tau = X * v/|v|$, where X is the term that is different in each case.

L159: Each simulation is run for 20 years => After optimizing, the sliding parameters are fixed and each simulation is additionally run for 20 years

L161: influence the => influence on the

L167: Please reiterate here which value of $M\_max$ is used for clarity.

$M\_max$ is in fact not used at all in this case, an important clarification to add.

L253: "higher values of m respond". <= please rephrase slightly as the values of m do not respond.

Figs. 7 & 8, captions: The contributions of each region => The contributions of the whole domain as well as of each region

L309: Kohler Glaciers, the same locations => Kohler Glaciers - the same locations